# Elucidation of the Anticancer Mechanism of Durian Fruit (*Durio zibethinus*) Pulp Extract in Human Leukemia (HL-60) Cancer Cells

**DOI:** 10.3390/nu15102417

**Published:** 2023-05-22

**Authors:** Mohamad Sitheek Abdul Rahman, Sivakumari Kanakarajan, Rajesh Selvaraj, Ashok Kamalanathan, Sabiha Fatima, Manal Abudawood, Nikhat J. Siddiqi, Humidah Alanazi, Bechan Sharma, Maria de Lourdes Pereira

**Affiliations:** 1Department of Zoology, Presidency College, Chennai 600005, India; 2Department of Clinical Laboratory Sciences, College of Applied Medical Sciences, King Saud University, Riyadh 11433, Saudi Arabia; 3Department of Biochemistry, College of Science, King Saud University, Riyadh 11495, Saudi Arabia; 4Department of Biochemistry, Faculty of Science, University of Allahabad, Allahabad 211002, India; 5Department of Medical Sciences, CICECO—Aveiro Institute of Materials, University of Aveiro, 3810-193 Aveiro, Portugal

**Keywords:** *Durio zibethinus*, anticancer activity, apoptosis, DNA fragmentation, cell cycle

## Abstract

Durian (*Durio zibethinus* L.) grows widely in Southeast Asia. The pulp of the durian fruit contains carbohydrates, proteins, lipids, fibers, various vitamins, minerals, and fatty acids. This study was carried out to elucidate the anticancer mechanism of action of the methanolic extract of the fruit of *Durio zibethinus* (*D. zibethinus*) on human leukemia (HL-60) cells. The methanolic extract of *D. zibethinus* fruits exhibited its anticancer effect on HL-60 cells by inducing DNA damage and apoptosis. The DNA damage was confirmed by comet and DNA fragmentation assays. The methanolic extract of *D. zibethinus* fruits has been shown to cause cell cycle arrest in HL-60 cells during the S phase and G2/M phase. Additionally, the methanolic extract caused induction of the apoptotic pathway in the HL-60 cell line. This was confirmed by increased expression in pro-apoptotic proteins, viz., Bax protein expression, and a substantial reduction (*p* < 0.001) in anti-apoptotic proteins, viz., Bcl-2 and Bcl-xL expressions. Therefore, this study confirms that the methanolic extract of *D. zibethinus* exerts its anticancer effects on the HL-60 cell line, causing cell cycle arrest and induction of apoptosis by an intrinsic mechanism.

## 1. Introduction

The synthesis of novel formulations of herbal medicines for the treatment of a variety of healthcare conditions is a boon for poor nations [1]. In addition to being widely accessible and reasonably priced, herbal medications offer relatively safe alternatives to conventional cancer therapies. Many medicinal plants are currently being researched for potential use as models for the development of new therapeutic agents or as innovative medicines [2]. Durian (*Durio zibethinus* L.), belonging to the Bombacaceae family, is grown in Malaysia, the Philippines, Indonesia, and Thailand. Durian fruit pulp is rich in carbohydrates but also contains proteins, lipids, fibers [3], vitamin C, B group vitamins, and minerals such as sodium, zinc, and calcium. The pulp is also a source of myristic acid, linoleic acid, stearic acid, and palmitic acid [4]. The medicinal properties of durian fruits include antidiabetic, anti-cancer, anti-cardiovascular [5], anti-obesity [6], and antioxidant [7] effects. The pulp of durian has been reported to alleviate symptoms of anxiety, stress, and depression [8]. Durian fruit extracts have been reported to reduce the growth of Calu-6 and SNU-601 cell lines [9] and inhibit proliferative activity in breast cancer cell lines [10].

Leukemia is a malignant tumor that affects the hematopoietic system. It causes about 30% of deaths in children [11]. Leukemia results from mutations in genes that regulate cell division, cell differentiation, and cell death. This causes the cells to shift from a normal to a malignant state [12]. Hematopoietic cancers develop due to uncontrolled growth and the accumulation of immature blasts due to blocking the progression of cells to the maturation stage. This causes a deficiency of normally functioning blood cells [13], leading to serious consequences. The cure for leukemia currently remains unsatisfactory, so natural products offer a more promising treatment option with fewer side effects. Many cancer cell lines, such as HL-60, NB-4, and Jutkat, have been used in vitro to screen the anticancer activity of compounds from natural sources [14]. The low toxicity of natural compounds has been widely used in the development of anticancer drugs. A previous study reported the anti-proliferative effect of durian fruits on HL-60 cells [15]. However, the mechanism involved in the anticancer and apoptotic properties of durian fruit has not been fully investigated. Hence, in this study, we aimed to investigate the anticancer mechanisms of durian fruit pulp extract using HL-60 cells in an in vitro setting.

## 2. Materials and Methods

### 2.1. Preparation of Fruit Extracts

Durian fruits (*Durio zibethinus*) were procured from the Koyambedu fruit market, Chennai, Tamil Nadu, India. The fruits were identified and authenticated by Professor P. Jayaraman of the Plant Anatomy and Research Centre, Tambaram, Chennai, India (No. PARC/2017/2122). The endocarps of the fruits were separated and dried in the shade for about two weeks. 50 g of dried endocarp were powdered and extracted with methanol solvent (1:10 *w*/*v*) for 72 h at room temperature. The extracts were filtered using Whatman No. 1 filter paper at 45–55 °C under decreasing pressure. The filtrates were concentrated (ROLEX) and used for further studies [16].

The solvent and the dose of extract were chosen based on previous studies [15]. The 48 h IC50 dose in methanol (21.56 µg/mL) was used in this study.

### 2.2. Cell Culture HL-60

HL-60 cell lines were procured from the National Centre for Cell Science, Pune, India. The cells were grown in a 75 cm^2^ culture flask containing RPMI-1640 supplemented with fetal bovine serum (10%), penicillin (50 U/mL), and l-glutamine (2 mM) at 37 °C with 5% CO_2_ in a humidified environment.

### 2.3. Cell Cycle Assay

The cell cycle was studied using propidium iodide (PI). Approximately 5–6 × 10^5^ Hl-60 cells were treated with the IC_50_ concentration of *D. zibethinus* methanol extract for 48 h. The treated cells were then harvested and transferred to tubes (2 mL). The cells were washed, then centrifuged (4 min) at 200× *g*, washed using 1 × PBS, and again centrifuged. Later, all samples were fixed using chilled ethanol (70%), followed by incubation (−20 °C for 3 h). After incubation, fixed cells were washed, centrifuged, and resuspended in 1× PBS buffer. This was followed by adding the cells to 50 μL RNase (Thermo-Fisher Scientific, Waltham, MA, USA) and incubating for 15 min. After 15 min, propidium iodide (50 μL; Santa-Cruz Biotechnology, Dallas, TX, USA) was added to the samples and kept for 20 min in the dark. Cell analysis was performed through the Beckman Vantage flow cytometer (Beckman Coulter Life Sciences, Indianapolis, IN, USA), and quantification was done using Summit 4.3 software (Beckman-Coulter, Inc., Brea, CA, USA)

### 2.4. Annexin V-FITC and PI Staining

After treatment with the *D. zibethinus* methanol extract, Annexin V-FITC/Propidium Iodide (PI) staining was performed to detect apoptosis in HL-60 cells using the previous method with some modification [17]. Briefly, the HL-60 cells were treated with the Durian methanol extract at IC50 concentrations for 48 h in a 96-well plate. This was followed by centrifugation at 400× *g* for 5 min at room temperature. The supernatant was carefully discarded, and the cells were washed using 1× Annexin binding buffer (100 µL) and again centrifuged. After carefully removing the supernatant, the suspended cells were incubated with 5 µL of Annexin V and 4 µL PI in the dark at room temperature for 15 min. After incubation, the cells were washed with 1× Annexin binding buffer, centrifuged at 2000× *g* rpm for 5 min at room temperature, and further resuspended in 1× Annexin binding buffer. The stained cells were examined under a fluorescent microscope (Zeiss, Oberkochen, Germany) to capture the stained cell images.

### 2.5. DNA Fragmentation Assay

HL-60 cells (3–4 × 10^5^) treated with or without the IC_50_ concentration of a methanol extract from durian pulp for 48 h were used to assess DNA fragmentation using DNA gel electrophoresis. A molecular weight DNA ladder was procured from Invitrogen^TM^ (Carlsbad, CA, USA). After the treatment, the cells were washed with PBS, and centrifuged at 200× g for five minutes to obtain the cell pellet, which was then re-suspended in 0.5 mL of lysis buffer and incubated for 30 min at 37 °C. The lysate was centrifuged at 14,000× *g* for ten minutes. Proteinase K (0.5 mg/mL) was added to the supernatant and incubated for 3 h at 50 °C. Further, the extraction of DNA was performed by a phenol, chloroform, and isoamyl alcohol (25:24:1) extraction procedure [18]. The pellet was centrifuged and air-dried at room temperature and further treated with RNAase A (100 µg/mL) for 60 min at 37 °C. The DNA fragments were separated by agarose gel electrophoresis (1.5%) containing ethidium bromide, and the bands were visualized under UV light.

### 2.6. Comet Assay

DNA was extracted from the untreated and treated cells and then utilized for the comet assay. The comet assay was carried out following the method of [19]. Slides prepared earlier were incubated in a cold lysis solution (which contained 0.1 M EDTA, 2.5 M NaOH, 0.01 M Tris, and 1% Triton X-100, pH 10). Slides were incubated in fresh alkaline electrophoresis buffer (300 mM NaOH, 1 mM EDTA at pH 13). Electrophoresis was carried out at 4 °C for 20 min at 25 V and 300 mA. Slides were neutralized by immersing them in Tris buffer (0.4 M, pH 7.5), followed by staining with ethidium bromide. The stained cells were analyzed under a fluorescence microscope (Zeiss, Oberkochen, Germany) and quantified by the CASP software 1.2.3 b1 (CASPLab, Wroclaw, Poland) program. The treatment was carried out in duplicate, and 50 randomly selected comets from each slide were analyzed to assess the degree of DNA damage, which was represented by the mean percentage of total DNA in the tail.

### 2.7. Western Blotting Analysis of Apoptotic Protein Expression

The expression of Bcl-2, Bcl-xL, Bax, Caspase-3, Caspase-9, and β-Actin was evaluated in untreated and 48 h IC_50_ concentration of *D. zibethinus* methanol extract-treated HL-60 cells. A mixture comprising 50 mg of cell lysate and 2× sample buffer was boiled for 5 min. The sample combination was electro-transferred to a PVDF membrane after running on 12% SDS-PAGE gels for 2 h at 100 V in 1× running gel buffer. A blocking buffer containing 10% non-fat dried milk was used to block the membrane overnight. Primary antibodies of Bcl-2 (1:1000; sc-7382) Bcl-xL (1:100; sc-8392), Bax (1:1000; sc-7480), Caspase-3 (1:1000; sc-7272), and Caspase-9 (1:1000; sc-56076) procured from Santa Cruz Biotechnology, Inc., Dallas, TX, USA were incubated on the membranes for 6 h together with a loading control, β-Actin (1:1000; sc-47778; Santa Cruz Biotechnology, Inc., Dallas, TX, USA). Membranes were rinsed three times with a blocking buffer for 10 min, each after being incubated with primary antibodies. Using a secondary antibody linked to horseradish peroxidase, membranes were incubated. The expression of the protein was detected by a pierce-enhanced chemiluminescence sensor (Thermo Fisher Scientific Inc., MA, USA), and using a GS-670 imaging densitometer (Bio-Rad, Hercules, CA, USA), the bands were quantified.

### 2.8. Statistical Analysis

The results of apoptotic studies were obtained in triplicate, and the mean ± S.E. of three individual experiments was calculated. Data were analyzed by two-way analysis of variance to assess the significance between the control and experimental groups. Statistical significance was considered at the *p* < 0.001 level.

## 3. Results

### 3.1. D. zibethinus Methanol Extracts Induced Cell Cycle Arrest in HL-60 Cells

The inhibition of cell survivability due to the induction of apoptosis in proliferating cancer cells can be achieved either by inhibition of some phase of the normal cell cycle or by the induction of extensive DNA damage that leads to DNA fragmentation. Thus, cell cycle analysis through flow cytometry was performed to detect its relationship to decreased viability. A cell cycle study was performed on HL-60 cells treated with *D. zibethinus* methanol extract at the IC_50_ concentration for 48 h (Figure 1). In contrast to untreated cells, treated cells showed a significant decrease in cell numbers in the G1 phase. This was associated with significant (*p* < 0.05) cell cycle arrest in the S phase and some cell arrest in the G2/M phase. Cell cycle analysis indicated that 48 h IC_50_ concentrations of *D. zibethinus* methanol extract-treated cells got arrested in the S phase and G2/M phase, indicating that the initiation event may have occurred in the S phase of the cell cycle.

### 3.2. D. zibethinus Methanol Extracts Induced Apoptosis in HL-60 Cells

Annexin V-FITC and PI staining were performed, as the initiation of apoptosis is one of the crucial factors responsible for cell survivability. Figure 2 shows pictures of the HL-60 cells stained with Annexin V-FITC, PI, and Annexin V-FITC/PI and seen under a confocal microscope after being treated with *D. zibethinus* methanol extract at IC_50_ 48 h concentration. Annexin V-FITC, PI, and Annexin V-FITC/PI did not stain the untreated HL-60 cells. In contrast, HL-60 cells treated with *D. zibethinus* methanol extract at the IC_50_ 48 h concentration displayed strong positive staining with Annexin V-FITC, PI, and Annexin V-FITC/PI. These findings demonstrate unequivocally that, as compared to untreated cells, *D. zibethinus* methanol extract at the IC_50_ 48 h concentration causes cell death even at the lowest concentration (Figure 2).

### 3.3. D. zibethinus Methanol Extracts Induced DNA Damage in HL-60 Cells

The cell cycle arrest at the G2/M phase indicates intracellular DNA damage. Analysis of DNA damage was performed using the comet assay and DNA fragmentation assays. Figure 3 shows the fluorescence microscope images of the HL-60 cell nuclei in untreated cells and cells treated with the IC_50_ 48 h concentration of *D. zibethinus* methanol extract. According to the comet assay results, compared to untreated HL-60 cells, the damage was significantly higher in treated cells. Comet-like tail cells represented the fragmented DNA that was moving out of the nucleus.

Following apoptosis, the DNA strand break was confirmed by DNA fragmentation assay to examine the apoptotic effect of the 48 h treatment of the IC_50_ concentration of *D. zibethinus* methanol extract on HL-60 cells. In the gel electrophoresis, using a DNA ladder as a comparison, DNA fragments were confirmed. In agreement with apoptosis, the results showed an apparent DNA breakage in HL-60 cells treated with *D. zibethinus* methanol compared to untreated cells (Figure 4).

### 3.4. Effect of D. zibethinus Extract Treatment on Apoptotic Protein Expression

DNA damage triggers Caspase activation and apoptosis through an alteration in the expression of Bcl-2 family proteins. Hence, using Western blot, the expression of anti-apoptotic Bcl-2 and Bcl-xL proteins as well as pro-apoptotic Bax protein was assessed in untreated and 48 h *D. zibethinus* extract-treated HL-60 cells. When compared to untreated HL-60 cells, the lysate from *D. zibethinus* extract-treated HL-60 cells showed a significant increase (*p* < 0.001) in pro-apoptotic Bax protein expression. However, *D. zibethinus* extract treatment in HL-60 cells induced a substantial reduction (*p* < 0.001) of anti-apoptotic Bcl-2 and Bcl-xL protein expressions (Figure 5A,B).

Further, the Western blot analysis of Caspase-9 and Caspase-3, key mediators of apoptosis, showed that, compared to the untreated cells, *D. zibethinus* extract treatment enhanced the levels of Caspase-9 and Caspase-3 in HL-60 cells (Figure 5A,C). Our results indicate the role of the altered Bcl family and activation of Caspase-3 and Caspase-9 as potential molecular mechanisms involved in the induction of *D. zibethinus* extract-induced apoptosis in HL-60 cells.

## 4. Discussion

In the recent past, there has been an increased interest in the search for natural sources of anticancer compounds. Therefore, various plants have been identified that contain anticancer compounds. Current treatments for cancer include chemotherapy and radiotherapy. Chemotherapy, however, remains one of the standard treatments, although it is associated with deleterious side effects. One of the problems in the treatment of cancer is the emergence of drug resistance in cancer cells [14]. Many anti-cancer drugs are obtained from natural sources or by derivatization of compounds from natural sources [20,21].

These compounds have the advantage of being effective at different phases of cancer and acting on multiple molecular targets, thereby activating multiple cellular pathways [22]. Although plant-derived compounds are the main sources of anticancer drugs, only a small fraction of them have been evaluated for their anticancer activity and mechanisms of action [14].

Over the years, different cell lines have been used as established models for screening the anticancer activity of compounds from natural sources [14]. These cell lines serve as models to evaluate the cytotoxic properties of anticancer compounds and their mechanisms of action [23]. Various leukemia cell lines (HL-60, U937, and THP-1) have been used as models for in vitro screening of compounds with anticancer activity. Earlier studies by [15] demonstrated the anticancer potential of durian fruit (*Durio zibethinus*) pulp extracts on HL-60 cell lines.

The cell cycle involves orchestrated events that result in the duplication of the cell’s genome, its growth, and finally cell division [24]. The order, integrity, and fidelity of the cell cycle are maintained by various checkpoints in the cell cycle [25]. These ensure cell size growth, the replication of DNA, chromosomal integrity, and segregation during mitosis. Thus, cell cycle arrest is a mechanism for the tumor cell to repair its damaged DNA. Therefore, the abrogation of cell cycle checkpoints before the DNA is repaired can activate apoptosis, causing cell death [26]. The present study indicates that the anticancer effect of *Durio zibethinus* fruit extract is exerted by activating apoptosis and arresting the cell cycle in the S phase and in the G2/M phase. This cell cycle arrest may have been triggered by mitochondrial dysfunction [27].

In the present study, *D. zibethinus* pulp extracts were shown to induce apoptosis, which was confirmed by Annexin V-FITC and PI staining. This induction of apoptosis may be attributed to the presence of flavonoids in the extracts [14,15].

Flavonoids show anticancer properties, both in vivo and in vitro, which also include antileukemic activities [11]. Inhibition of the proliferation of cancer cells is the consequence of stimulation of apoptosis and disruption of the cell cycle [28]. Apoptosis is a coordinated process involving the synergistic and opposing actions of several oncogenes and oncoproteins. It can be triggered by an intrinsic pathway caused by mitochondrial membrane permeation or an extrinsic pathway, which is the binding of death receptors to their ligands. Induction of the apoptotic pathway is one of the strategies for cancer treatment. In this study, the methanolic extract of *D. zibethinus* fruit pulp appeared to trigger apoptosis by triggering the intrinsic pathway. The intrinsic apoptotic pathway involves the activation of Caspase-9, which, in turn, activates caspase 3. Therefore, Western blotting was carried out to confirm the activation of caspases 9 and 3. The methanolic extract of *D. zibethinus* fruit pulp caused a significant enhancement in Caspase-9 and -3, along with the Bax protein, which is a pro-apoptotic protein. This confirmed that the methanolic extract of *D. zibethinus* fruit pulp caused the induction of an intrinsic apoptotic pathway in HL-60 cells. On the other hand, treatment with *D. zibethinus* extract in HL-60 cells caused a significant reduction (*p* < 0.001) in the expression of anti-apoptotic Bcl-2 and Bcl-xL proteins. The induction of apoptosis by the fruit extracts of *D. zibethinus* may be attributed to their polyphenol content [15]. Polyphenols have been reported to induce apoptosis and inhibit the growth of cancer cells [29,30,31]. DNA fragmentation occurs during the late stages of apoptosis [32]. DNA fragmentation is caused by the activation of cellular nucleases, which are activated during apoptosis [33]. In this study, the methanolic extract of *D. zibethinus* fruit pulp caused apoptosis in HL-60 cells. Apoptosis-inducing factor I (AIF I) is a protein present in the inter-mitochondrial space. Apoptosis results in the proteolysis of AIF I. AIF is then translocated to the nucleus, where it causes chromatin condensation and DNA degradation in a Caspase-independent manner [34]. In this study, the loss of mitochondrial membrane potential may be responsible for the release of AIF I, which, in turn, would cause DNA damage.

## 5. Conclusions

This study concludes that the anticancer activity of the methanolic extract of *D. zibethinus* fruits is exerted by initiating DNA breaks and causing cell cycle arrest leading to apoptosis in HL-60 cell lines.

## Figures and Tables

**Figure 1 nutrients-15-02417-f001:**
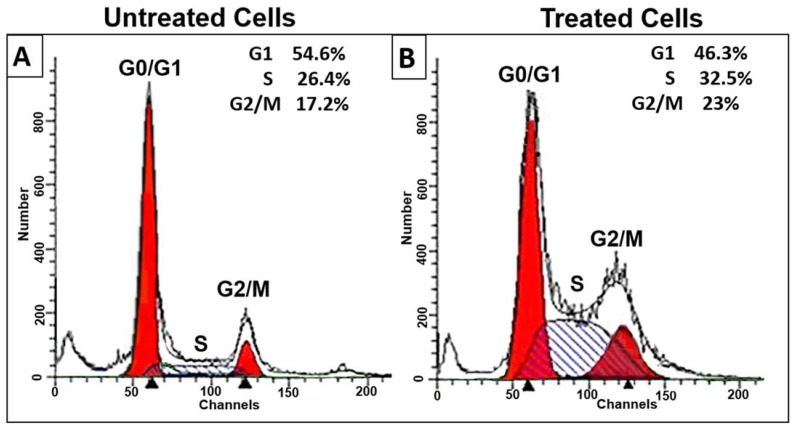
Effect of 48 h of *D. zibethinus* methanol extract treatment on HL-60 cells. The cell cycle distribution assessed by flow cytometry in (**A**) untreated and (**B**) *D. zibethinus* methanol extract treatment on HL-60 cells shows significant cell accumulation in the S phase and G2/M phase. (**C**) Bar graph showing the accumulation of HL-60 cells in each cell cycle phase after 48 h of *D. zibethinus* methanol extract treatment. Values are the mean ± S.E. of three individual experiments. * Values are statistically significant at *p* < 0.05 relative to the control. ** Values are statistically significant at *p* < 0.001 relative to the control.

**Figure 2 nutrients-15-02417-f002:**
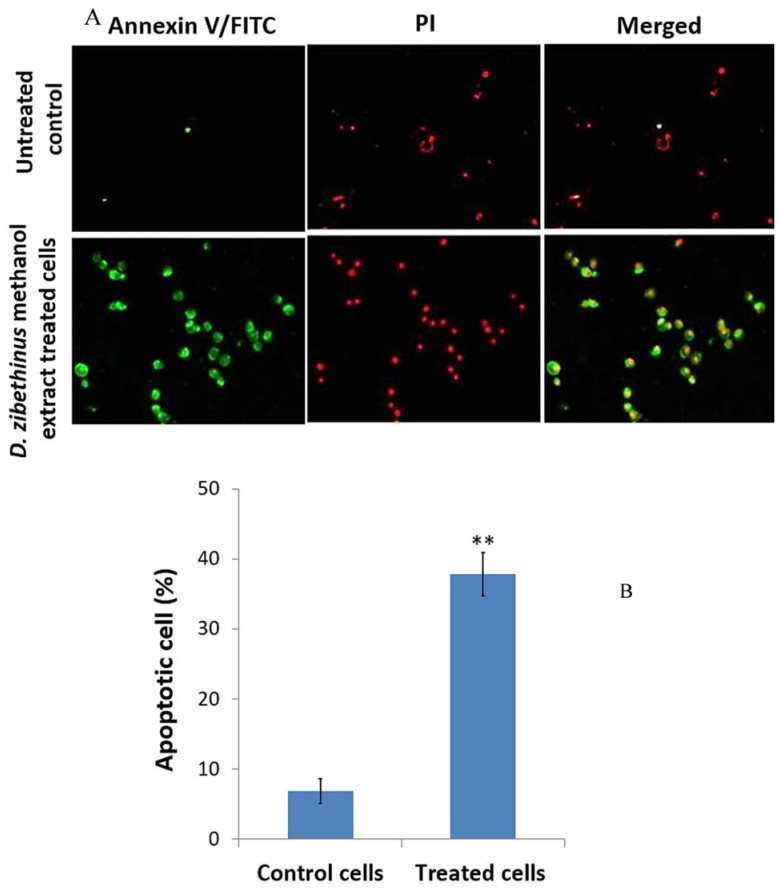
Evaluation of apoptosis using Annexin-V FITC/ Propidium Iodide (PI) staining. (**A**) Images of the apoptotic cells (green color), necrotic cells (red color), and later stages of apoptotic phase (merged) of 48 h IC_50_ concentrations of *D. zibethinus* methanol extract-treated and untreated control HL-60 cells. (**B**) Statistical analysis of the staining results indicates a significant increase in the apoptosis rate of HL-60 cells treated with *D. zibethinus* methanol extract compared with the untreated control group. Values are the mean ± S.E. of three individual experiments. ** Values are statistically significant at *p* < 0.001 relative to the control.

**Figure 3 nutrients-15-02417-f003:**
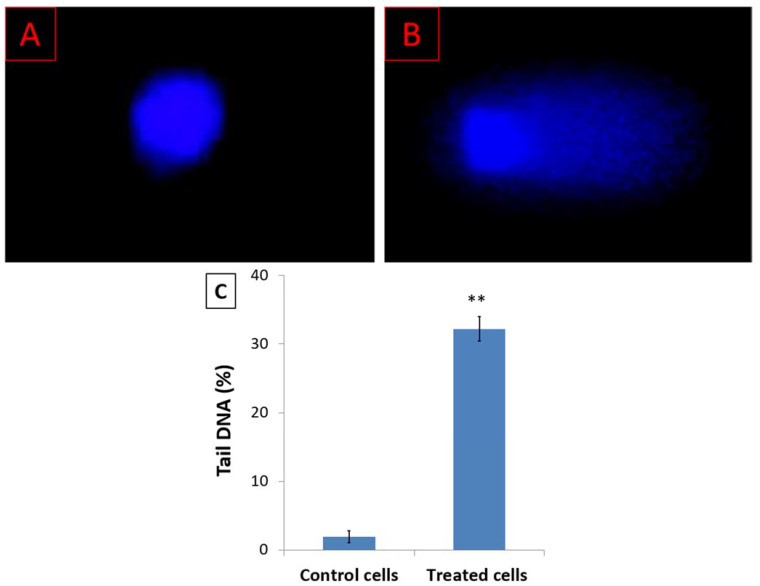
Analysis of DNA damage by comet assay for (**A**) untreated HL-60 cells and (**B**) IC50 48 h concentration of *D. zibethinus* methanol extract-treated HL-60 cells; (**C**) The quantitative analysis of the comet assay results indicates a significant increase in DNA damage of HL-60 cells treated with *D. zibethinus* methanol extract compared with the untreated control group. Values are the mean ± S.E. of three individual experiments. ** Values are statistically significant at *p* < 0.001 relative to the control.

**Figure 4 nutrients-15-02417-f004:**
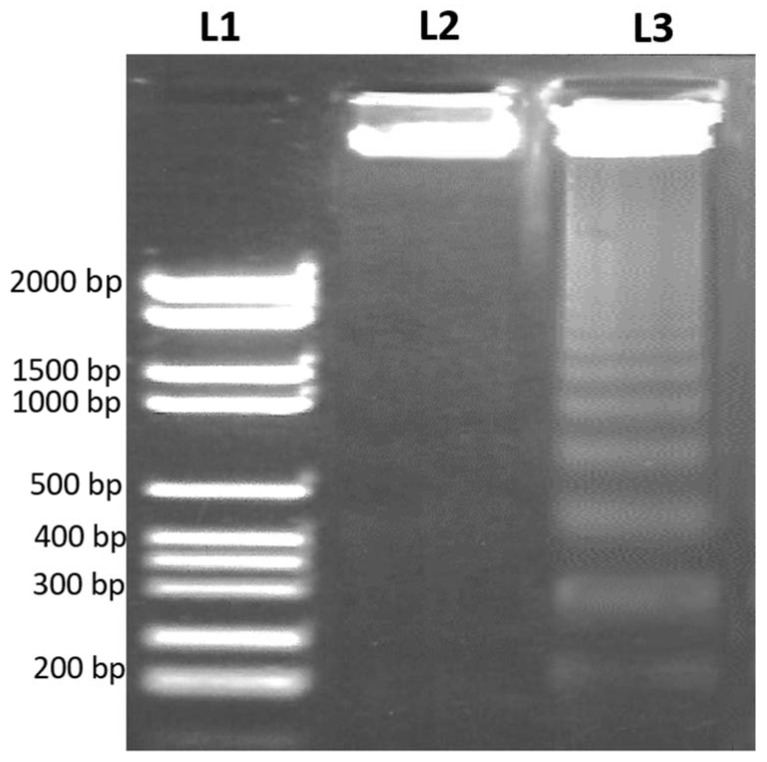
DNA fragmentation assay by agarose gel electrophoresis to examine apoptosis. (**L1**) DNA ladder (2000 bp); (**L2**) Untreated control HL-60 cells; (**L3**) *D. zibethinus* methanol extract-treated HL-60 cells. The results are the representation of figures from three independent experiments.

**Figure 5 nutrients-15-02417-f005:**
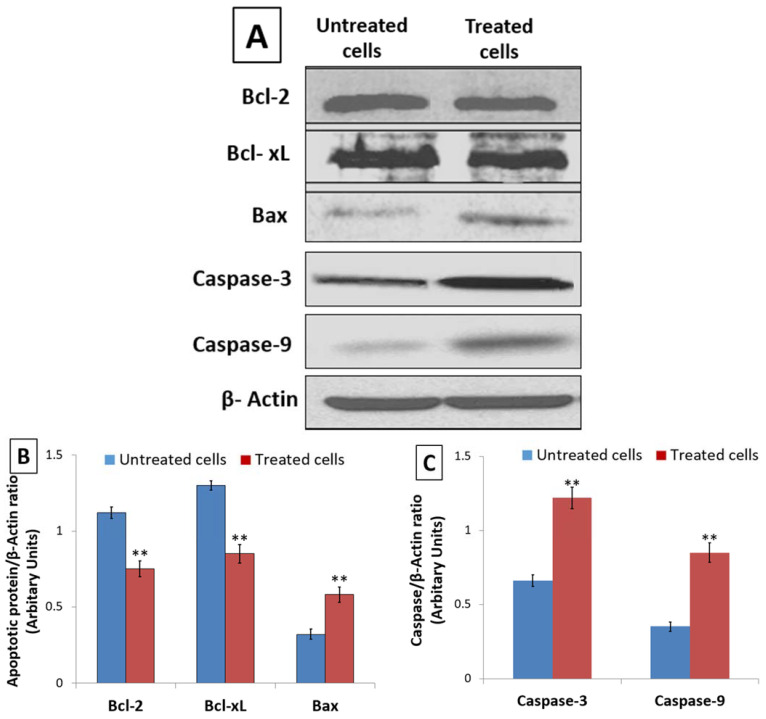
Effect of 48 h of treatment on the apoptotic marker expression. (**A**) Blot bands of protein expression of Bcl-2, Bcl-xL, Bax, Caspase-9, and Caspase-3 in untreated and *D. zibethinus* extract-treated HL-60 cells. The protein density of (**B**) Bcl-2, Bcl-xL and Bax, and (**C**) Caspase-9, and Caspase-3 were quantified relative to β-Actin, used as an internal control. Values are mean ± S.E. of three individual experiments. ** Values are statistically significant at *p* < 0.001 relative to the control.

## Data Availability

The corresponding author will provide data from the current study upon reasonable request.

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
