# Peer review of "Elucidation of the Anticancer Mechanism of Durian Fruit (Durio zibethinus) Pulp Extract in Human Leukemia (HL-60) Cancer Cells"

_nutrients, 2023, doi:10.3390/nu15102417_

Round 1
Reviewer 1 Report (Previous Reviewer 3)
The manuscript has been slightly revised and still needs to be modified.
Major comment:
Further studies are needed to identify the active compounds responsible for the action of durian extract.
Minor comments (see pdf file attached):
Introduction: The authors did not emphasize the novelty and importance of the research conducted
The Discussion section is not a discussion but a description of the results. Therefore, these two chapters (Results and Discussion) should be combined into one.
Conclusions: this section needs to be modified as it does not emphasize the importance of the research carried out, and even reduces its value
The manuscript contains numerous typographical errors. Text formatting should be carefully checked and corrected.
References are incorrectly quoted in the text of the manuscript.
References should be adapted to the journal's requirements. In the References section, abbreviations of journal names should be provided. Some references are invalid.
The language should be modified carefully.

Author Response
The manuscript has been slightly revised and still needs to be modified.
Major comment:
Response - We are grateful to the reviewers for their valuable comments which helped us in improving the quality of our manuscript. All the suggested changes and corrections are highlighted in the manuscript.
Further studies are needed to identify the active compounds responsible for the action of durian extract.
Response - Indeed, further studies will be carried out to identify the active compounds responsible for the action of durian extract. This has been planned and will be done in future.
Minor comments (see pdf file attached):
Introduction: The authors did not emphasize the novelty and importance of the research conducted
Response – A paragraph has been added on “Leukemia” in the introduction and its relevance to the present work has been mentioned.
The Discussion section is not a discussion but a description of the results. Therefore, these two chapters (Results and Discussion) should be combined into one.
Response – We don’t know if the journal rules permit the merging of results and discussion. But the discussion has been elaborated.
Conclusions: this section needs to be modified as it does not emphasize the importance of the research carried out, and even reduces its value.
Response – The conclusion has been rewritten in a clear manner.
The manuscript contains numerous typographical errors. Text formatting should be carefully checked and corrected.
Response – The typographical errors have been corrected.
References are incorrectly quoted in the text of the manuscript.
Response- The references in the text have been matched with the list of references.
References should be adapted to the journal's requirements. In the References section, abbreviations of journal names should be provided. Some references are invalid.
Response- The journal at this stage accepts free format submission. Once the paper is accepted and they ask us to format, we will format as per journal requirements.
The language should be modified carefully.
Response- The language has been modified.

Reviewer 2 Report (Previous Reviewer 2)
The resubmitted manuscript “Elucidation of Anticancer Mechanism of Durian Fruit (Durio zibethinus) Pulp Extract on Human Leukemia (HL-60) Cancer Cells” by Mohamad Sitheek at al. presents results on the anticancer and apoptosis inducing activities of methanol extract of D. zibethinus fruit against human leukemia HL-60 cell line. The authors evaluated the potential of the methanol extract to affect the cell cycle distribution and induce apoptosis of HL-60 cells. Additionally, the authors tested if the methanol extract can induce DNA damage.
Frankly, when comparing the original submission and the current version of the manuscript, I don´t see any significant improvement, although rejection or major revision were requested.
Overall, the findings of this study are still preliminary and experiments were not performed properly. It is unacceptable that only single 21.56 µg/ml dose of methanol extract was tested and no time-dependent experiments were performed or positive controls were included.
Comments and recommendations that were not resolved:
1. Figure 2: The time-dependent effect of various concentrations of tested extract on cell cycle of HL-60 cells must be tested.
2. Figure 3 and Table 1: How many cells were analysed in comet assay? Statistical analysis is mandatory.
3. Figure 4: Picture is not of good quality. More concentrations must be tested and positive control for apoptotic DNA fragmentation must be included. Have been the samples treated with RNase A before purification of DNA (check the methodology)?
4. Figure 5: The time-dependent effect of various concentrations of tested extract on particular markers of apoptosis must be provided.
5. Methodology related to Cell cycle assay: Why the cells were trypsinized? HL-60 cells grow in suspension!
6. English needs extensive editing and revision.
Author Response
The resubmitted manuscript “Elucidation of Anticancer Mechanism of Durian Fruit (Durio zibethinus) Pulp Extract on Human Leukemia (HL-60) Cancer Cells” by Mohamad Sitheek at al. presents results on the anticancer and apoptosis inducing activities of methanol extract of D. zibethinus fruit against human leukemia HL-60 cell line. The authors evaluated the potential of the methanol extract to affect the cell cycle distribution and induce apoptosis of HL-60 cells. Additionally, the authors tested if the methanol extract can induce DNA damage.
Frankly, when comparing the original submission and the current version of the manuscript, I don´t see any significant improvement, although rejection or major revision were requested.
Overall, the findings of this study are still preliminary and experiments were not performed properly. It is unacceptable that only single 21.56 µg/ml dose of methanol extract was tested and no time-dependent experiments were performed or positive controls were included.
Comments and recommendations that were not resolved:
- Figure 2: The time-dependent effect of various concentrations of tested extract on cell cycle of HL-60 cells must be tested.
Response - Ans- In our previous study, the percent cell viability of HL-60 cells treated with various extracts (chloroform, hexane, ethyl acetate, and methanol) of D. zibethinus at different concentrations (0, 25, 50, 75, 100, and 125 μg/ml) for 24 h and 48 h was assessed. The result of the preliminary study showed that a 48 h IC50 concentration of 21.56 µg/ml of D. zibethinus methanol extract profoundly controlled HL-60 cell proliferation. Based on this 48 h IC50 concentration of D. zibethinus methanol extract was taken for further studies.
Reference
ANTI-PROLIFERATIVE EFFECT OF DURIAN FRUIT (DURIO ZIBETHINUS) AGAINST HL-60 CELLS AND ITS PHYTOCHEMICAL ANALYSIS
Link - https://www.researchgate.net/publication/343365671
- Figure 3 and Table 1: How many cells were analysed in comet assay? Statistical analysis is mandatory.
Response- The number of cells analyzed are added and highlighted in the Materials and Methods- section 2.6
The treatment was carried out in duplicate and 50 randomly selected comets from each slide were analyzed to assess the degree of DNA damage. and represented by the mean percentage of total DNA in the tail.
Table 1 is removed as DNA damage is represented by the mean percentage of total DNA in the tail (Figure 3).
Please check the legend of Figure 3 for the statistical analysis.
- Figure 4: Picture is not of good quality. More concentrations must be tested and positive control for apoptotic DNA fragmentation must be included. Have been the samples treated with RNase A before purification of DNA (check the methodology)?
Response- Figure 4 has been changed to improve the quality of the figure and the positive control for apoptotic DNA fragmentation is added.
- Figure 5: The time-dependent effect of various concentrations of tested extract on particular markers of apoptosis must be provided.
Response- In our preliminary study HL-60 cells were treated with various extracts (chloroform, hexane, ethyl acetate, and methanol) of D. zibethinus at different concentrations (0, 25, 50, 75, 100, and 125 μg/ml) for 24 h and 48 h. The result of the preliminary study showed that a 48 h IC50 concentration of 21.56 µg/ml of D. zibethinus methanol extract profoundly controlled HL-60 cell proliferation. Based on this 48 h IC50 concentration of D. zibethinus methanol extract was taken for further studies.
Reference
ANTI-PROLIFERATIVE EFFECT OF DURIAN FRUIT (DURIO ZIBETHINUS) AGAINST HL-60 CELLS AND ITS PHYTOCHEMICAL ANALYSIS
Link - https://www.researchgate.net/publication/343365671
- Methodology related to Cell cycle assay: Why the cells were trypsinized? HL-60 cells grow in suspension!
Response -The HL-60 cells were not trypsinized, the writing mistake is corrected in the Methodology related to the Cell cycle assay.
- English needs extensive editing and revision.
Response -The English has been edited by one of the coauthors viz., Prof. Maria.

Round 2
Reviewer 1 Report (Previous Reviewer 3)
I have no additional comments, except that Figure 5A is illegible.
Author Response
Figure 5A has been added in the manuscript and also a separate file is being submitted.

Reviewer 2 Report (Previous Reviewer 2)
I have no further comments and suggestions.
Author Response
There are no comments.
This manuscript is a resubmission of an earlier submission. The following is a list of the peer review reports and author responses from that submission.
Round 1
Reviewer 1 Report
Dear authors:
The manuscript entitled "Anticancer and Induced Apoptotic Evaluation of Durian Fruit (Durio zebethinus) Pulp Extracts on Human Leukemia (HL-60) Cancer Cells":
1. Needs extensive editing of English language.
2. This manuscript does not support enough data to show the apoptotic effects of Durio zebethinus extracts on HL-60. Moreover, the data presentations should be extremely improved by using the graphs and pictures in higher resolutions. For most of the data, the quantifications have not been done or presented such as mitochondrial membrane potential assay, western blot assays, and DNA damage.
3. Statistics are missing: P value, percentage of effects, graph bars, etc.
Line 282, 399: What does membrane cells mean?
Line 415: Did you measure ROS? If so, why the data were not presented here?
Author Response
The manuscript entitled "Anticancer and Induced Apoptotic Evaluation of Durian Fruit (Durio zebethinus) Pulp Extracts on Human Leukemia (HL-60) Cancer Cells":
- Needs extensive editing of English language.
The manuscript has been rewritten and edited for English language.
- This manuscript does not support enough data to show the apoptotic effects of Durio zebethinus extracts on HL-60. Moreover, the data presentations should be extremely improved by using the graphs and pictures in higher resolutions. For most of the data, the quantifications have not been done or presented such as mitochondrial membrane potential assay, western blot assays, and DNA damage.
The graph and picture quality has been improved. Quantification of most of the data has been done and included as graph with significance values.
- Statistics are missing: P value, percentage of effects, graph bars, etc.
Statistics have been added and P values have been introduced in the graphs.
- Line 282, 399: What does membrane cells mean?
The above lines have been deleted.
- Line 415: Did you measure ROS? If so, why the data were not presented here?
ROS was not measured.

Reviewer 2 Report
The manuscript “Anticancer and Induced Apoptotic Evaluation of Durian Fruit (Durio zebethinus) Pulp Extracts on Human Leukemia (HL-60) Cancer Cells” by Mohamad at al. presents results on the anticancer and apoptosis inducing activities of methanol extract of Durio zebethinus against human leukemia HL-60 cell line. The authors evaluated the activity of the extract by MTT, comet and DNA fragmentation assays, by flow cytometry, tunnel assay, and western blotting.
Overall, the findings of this study are preliminary and experiments were not performed properly. Only single concentration of extract was tested (IC50) and no positive controls were included.
Some other points and suggestions are included below.
Comments and recommendations:
1. The analysis of composition of chloroform, ethyl acetate, hexane, and methanol extracts of Durio zebethinus must be executed and data must be included in the manuscript.
2. The IC50 values need to be presented in Table as part of the Result section, not Methodology. Values should include also statistical analysis (e.g. S.D.)
3. Figure 1: The authors stated that “Figure 1 shows pictures of the HL-60 cells stained with Annexin V-FITC, PI, and Annexin V-FITC/PI and seen under a confocal microscope after being treated with D. zebethinus methanol extract at IC50 48 h concentration.”. Actually, the cells were probably stained with Annexin V-FITC/PI kit and the third column represent merged pictures. The methodology of “Annexin V-FITC and PI staining” is also confusing as no PI staining step is mentioned. It needs to be reformulated.
4. Figure 2: This is unusual presentation of data from the Comet assay. The figure showing more comets need to be provided for untreated and treated HL-60 cells. Include some positive control (e.g. H2O2).
5. Table 1: Statistical analysis is needed. Indicate, how many cells/comets were analyzed.
6. Figure 3: Picture with better resolution for DNA fragmentation/DNA ladder must be provided. Also test the wider range of concentrations (including the one that inhibit growth for 80%). Include the positive control (e.g. 6 microM cisplatin for 24 hrs).
7. Figure 4, Table 2: The concentration and time-dependent effects of extract on cell cycle profile of HL-60 cells need to provided, including the statistical analysis.
8. Figure 5: The Figure has unacceptable quality.
9. Figure 6: The changed in mitochondrial membrane potential of HL-60 cells should be analyzed by flow cytometry.
10. The concentration and time-dependent wester blot analysis of impact of extract on particular markers of apoptosis must be provided.
Author Response
Reviewer # 2
Comments and recommendations:
- The analysis of composition of chloroform, ethyl acetate, hexane, and methanol extracts of Durio zebethinus must be executed and data must be included in the manuscript.
The analysis composition of chloroform, ethyl acetate, hexane and methanol has been published earlier which ahs been mentioned in the “Materials and methods” section and in the references.
Mohamad Sitheek A., Sivakumari K., Rajesh S., Ashok K. Anti-Proliferative Effect of Durian fruit (Durio zibenthinus) Against HL-60 cells and its phytochemical analysis. Journal of Advanced Scientific Research. 2020; 11 Suppl 5: 174-181.
- The IC50 values need to be presented in Table as part of the Result section, not Methodology. Values should include also statistical analysis (e.g. S.D.)
The IC 50 values have been removed and earlier reference has been quoted.
- Figure 1: The authors stated that “Figure 1 shows pictures of the HL-60 cells stained with Annexin V-FITC, PI, and Annexin V-FITC/PI and seen under a confocal microscope after being treated with D. zebethinus methanol extract at IC50 48 h concentration.”. Actually, the cells were probably stained with Annexin V-FITC/PI kit and the third column represent merged pictures. The methodology of “Annexin V-FITC and PI staining” is also confusing as no PI staining step is mentioned. It needs to be reformulated.
The methodology has been reformulated.
- Figure 2: This is unusual presentation of data from the Comet assay. The figure showing more comets need to be provided for untreated and treated HL-60 cells. Include some positive control (e.g. H2O2).
The figure has been improved and graph has been included in figure 3. The addition of positive control at this stage is beyond the scope of this study.
- Table 1: Statistical analysis is needed. Indicate, how many cells/comets were analyzed.
This could not be done.
- Figure 3: Picture with better resolution for DNA fragmentation/DNA ladder must be provided. Also test the wider range of concentrations (including the one that inhibit growth for 80%). Include the positive control (e.g. 6 microM cisplatin for 24 hrs).
The resolution of the figure has been improved. The addition of positive control at this stage is beyond the scope of this study.
- Figure 4, Table 2: The concentration and time-dependent effects of extract on cell cycle profile of HL-60 cells need to provided, including the statistical analysis.
This has been provided in Figure 2.
- Figure 5: The Figure has unacceptable quality.
The figure quality has been improved.
- Figure 6: The changed in mitochondrial membrane potential of HL-60 cells should be analyzed by flow cytometry.
This has been removed due to poor quality.
- The concentration and time-dependent wester blot analysis of impact of extract on particular markers of apoptosis must be provided.
This has been provided in figure 5.

Reviewer 3 Report
This manuscript does require some modifications, which I have outlined in the comments below:
The Latin name for durian is misspelled in the title of the manuscript.
The Abstract needs to be corrected - the first five sentences are more of an Introduction than an Abstract.
Keywords should be changed, e.g. durian fruit, anticancer activity, apoptosis, ...etc.
References are incorrectly quoted in the text of the manuscript.
Lines 84-92: this paragraph should be transferred to the Result section.
Lines 313-323: this is not a discussion.
The manuscript contains numerous typographical errors. Text formatting should be carefully checked and corrected.
References should be adapted to the journal's requirements. In the References section, abbreviations of journal names should be provided. Some references are invalid.
Further studies are needed to identify the active compounds responsible for the action of durian extract.
Author Response
Reviewer # 3
This manuscript does require some modifications, which I have outlined in the comments below:
The Latin name for durian is misspelled in the title of the manuscript.
This has been corrected.
The Abstract needs to be corrected - the first five sentences are more of an Introduction than an Abstract.
The abstract has been rewritten.
Keywords should be changed, e.g. durian fruit, anticancer activity, apoptosis, ...etc.
The above-mentioned key words have been included.
References are incorrectly quoted in the text of the manuscript.
They have been corrected.
Lines 84-92: this paragraph should be transferred to the Result section.
It has been rewritten.
Lines 313-323: this is not a discussion.
The whole discussion has been rewritten.
The manuscript contains numerous typographical errors. Text formatting should be carefully checked and corrected.
This has been done.
References should be adapted to the journal's requirements. In the References section, abbreviations of journal names should be provided. Some references are invalid.
The references have been reformatted.
Further studies are needed to identify the active compounds responsible for the action of durian extract.
This will be done in future studies.
